# OpenReview forum: "Memory Injection Attacks on LLM Agents via Query-Only Interaction"
_NeurIPS.cc/2025/Conference — NeurIPS 2025 poster_

### Official Review · Reviewer_yqoc · 2025-06-06

**Clarity:** 2
**Significance:** 4
**Originality:** 3
**Rating:** 5
**Confidence:** 3

**Summary:**

The authors proposed an memory injection attack against LLM agents where the attacker injects
 malicious records into the memory bank by only interacting with the agent via queries and output observations.
The authors introduced a sequence of bridging steps to link victim queries to the malicious reasoning steps.

**Questions:**

What is the major technical difference between filling the logical gap by bridging steps with [A]?
[A] PANDORA: Detailed LLM Jailbreaking via Collaborated Phishing Agents with Decomposed Reasoning

Why do the authors only explore reasoning-based agents?

Does the attack leverage specific properties of reasoning models?

Is the attack method generalizable to non-reasoning model-based agents? How does it perform and transfer?

How does the method perform on other reasoning models except GPT4 and GPT4o, e.g., DeepSeek.

How does the method perform on small-scale distilled models, e.g., DeepSeek-R1-Distill-Qwen-14B? Does model scale generally impact the performance of your attack?

Is it possible to design potential strategies to defend this attack?

**Ethical Concerns:**

["NO or VERY MINOR ethics concerns only"]

**Final Justification:**

The authors rebuttal clarified and addressed my concerns on whether the model needs to be a reasoning llm, the model size, and difference with related work, etc.

**Limitations:**

The paper also lacks a discussion of whether it is possible for agent builders to defend against this attack. Will defending these attacks hinder the agent's performance?

**Quality:**

4

**Strengths And Weaknesses:**

Pros:
The agent memory injection attack problem is very intriguing. The authors pointed out and explored a novel and practical setting, compared with existing work AgentPoison, which assumed the attacker can directly manipulate the memory bank, which is not possible in practice. From my opinion, formulating the problem from this perspective is highly novel and presenting techniques to resolve this problem is a large step-forward on making attacks to agents more close to real-world. Studying agent attacks under this setting will make the public more aware of attacks against agents and LLMs, largely benefiting the LLM security community.

The method designed based on bridging steps is also neat.

Cons:

The idea of attacking based on bridging steps does not seem completely new. For example, in [A], the authors explored decomposing jailbreak questions into smaller reasoning steps and making the attack less easier to be detected.

[A] PANDORA: Detailed LLM Jailbreaking via Collaborated Phishing Agents with Decomposed Reasoning

In general, the evaluation is less comprehensive.

A lot of existing agents are not based on reasoning models but based on normal LLMs. Through reading the paper, I feel unsured about whether the attacker's techniques heavily rely on reasoning LLMs, or they are also transferable to normal LLMs like Llama-2. Thus, it is less justified to explore only reasoning-based agents.

The involved LLMs in the evaluation is limited to GPT4 and GPT4o. However, there are many existing reasoning-based LLMs, e.g., DeepSeek.

It is also unclear how the model performs on smaller models, at the scale of 7B or 14B.

---

> ### Author Rebuttal · Authors · 2025-07-31
>
> Thank you for your thoughtful review! We sincerely appreciate your recognition of our realistic problem setting and bridging-based design, as well as your support for its impact on advancing agent and LLM security research.
>
> **W1, Q1: The use of bridging or decomposed reasoning steps shares similarities with prior work (e.g., *PANDORA*).**
>
> **R1:** We thank the reviewer for pointing out the resemblance to prior reasoning-based attacks such as *PANDORA*. While both involve multi-step reasoning, MINJA differs fundamentally in its motivation, threat model, and attack mechanism as shown in the table below.
>
> | **Aspect**| **PANDORA**| **MINJA (Bridging + PSS)**|
> |-|-|-|
> | **Attack Goal**| Jailbreak via decomposing adversarial prompts into stealthy sub-queries | Memory poisoning to influence future benign queries|
> | **Attack Form**| One query decomposed into multiple legal instructions (n -> {1, 1, ..., 1})          | One query with indication prompt and progressive shortening (n -> {n-1, n-2, ..., 1})|
> | **Decomposition Purpose & Technique**| Dilute harmful intent via task decomposition and agent collaboration | Bridge victim-target semantics and improve retrievability via progressive shortening (PSS) |
> | **Long-Term Effect**  | No, effective only within current dialogue| Yes, lasting impact via memory injection|
> | **Target System**| LLM-based collaborative agents| Memory-augmented LLM agents|
>
> Specifically, *PANDORA* focuses on immediate jailbreaks through multi-agent collaboration and task decomposition, aiming to bypass content filters in a single session. In contrast, *MINJA* targets long-term memory by injecting malicious records that logically bridge victim and target concepts, using a Progressive Shortening Strategy (PSS) to gradually reduce prompt cues. This enables the attack to persist in memory and influence future benign queries in a single-agent setting.
>
> We will add this comparison to the related work section of our revised paper. Thank you again for your comment.
>
> **W2, W3, W4; Q2, Q3, Q4, Q5, Q6:  Does the attack rely on the reasoning capabilities of LLMs? Would it still be effective on non-reasoning agents based on general-purpose LLMs like Llama-2?; Can the attack transfer to other reasoning models beyond GPT-4(o), such as DeepSeek?; How does model size affect the attack’s performance? Can the method generalize to smaller models like 7B or 14B?**
>
> **R2:** We thank the reviewer for the questions about the models used in the agent.
>
> First, we would like to clarify that our attack **does not require the LLM model to be a dedicated “reasoning model”**, but we leverage the reasoning abilities of LLMs (even the non-reasoning models can perform Chain-of-thought reasoning) in the agent. Reasoning is commonly used by agents [1][2] operating in complex, real-world scenarios—such as decision making, planning, and multi-step tool use. Therefore, we do not require specific properties of reasoning models.
>
> Second, we extend our evaluation to  ***DeepSeek-R1*** for broader evaluation. On QA Agent (for arbitrarily selected victim-target pairs 1, 4, and 7), DeepSeek-R1 achieves consistently high Injection and Attack Success Rates, closely matching GPT-4’s performance:
>
> ***DeepSeek-R1:***
> | **Victim-target Pairs**| **Injection Success Rate**| **Attack Success Rate**|
> |-|-|-|
> | Pair 1 | 1.0 | 1.0 |
> | Pair 4 | 1.0 | 1.0 |
> | Pair 7 | 1.0 | 0.9 |
>
> Third, we investigate the effect of model scale using ***Llama-2-7B***. However, this model only achieves **19.3% and 17.1% task accuracy** for pairs 1 and 7, respectively, even lower than random guessing on MMLU with four answer choices (25%). These results indicate that Llama2-2-7B fails to perform the base task reliably, such that it will not likely be adopted as the core model of the agent, making attack evaluation less meaningful. Therefore, regarding small-scale models, **a minimum level of base task competence is required for evaluating attacks**.
>
> We will complete the full set of experiments on DeepSeek-R1 and incorporate more discussion on these models in the revised paper.
>
> **Q7: Is it possible to design potential strategies to defend against this attack?**
>
> **R3:** Thank you for your interest in potential defense!
>
> As discussed in Section 5.4, MINJA poses a challenge to conventional defenses due to its query-only interaction mode and the plausibility of the injected prompts.
>
> While we discuss and explore several strategies—adversarial training, embedding-level memory sanitization, and prompt-level detection—each has notable limitations. For example, adversarial training suffers from scalability issues; embedding-level filtering struggles due to entanglement of benign and malicious records; and prompt-level detection, while promising, either lacks generalization (targeted prompts) or introduces high false positives (general prompts).
>
> These findings suggest that MINJA remains difficult to defend using existing mechanisms. We believe this highlights the need for future research into robust, low-cost defenses specifically tailored to memory poisoning attacks.
>
> [1] ReAct: Synergizing Reasoning and Acting in Language Models
>
> [2] Reflexion: Language Agents with Verbal Reinforcement Learning

---

> > ### Comment · Reviewer_yqoc · 2025-08-01
> >
> > I think the rebuttal is very successful and fully clarifies and addresses my concerns.

---

> > > ### Author Response · Authors · 2025-08-01
> > > **Thank you for your supportive comment**
> > >
> > > We’re truly delighted to hear that our rebuttal fully addressed your concerns. Your thoughtful feedback has been immensely helpful throughout the review process, and we sincerely appreciate your time and support!

---

### Official Review · Reviewer_KhxJ · 2025-06-25

**Clarity:** 3
**Significance:** 3
**Originality:** 3
**Rating:** 5
**Confidence:** 3

**Summary:**

This paper presents MINJA, a novel memory injection attack against large language model agents. Unlike previous work, it relies solely on query-based interaction to inject malicious records that can later influence the agent’s behavior during other users’ interactions. The authors introduce techniques like bridging steps, an indication prompt, and a progressive shortening strategy to craft effective injections. The experiments across multiple agents (e.g., healthcare, QA, shopping) show high success rates in both injection and downstream attacks, while preserving the agent’s original utility.

**Questions:**

Please answer how you may address the weaknesses above.

**Ethical Concerns:**

["NO or VERY MINOR ethics concerns only"]

**Final Justification:**

The authors have addressed my concerns about attack assumptions and practicality.

**Limitations:**

yes

**Quality:**

3

**Strengths And Weaknesses:**

Strengths:
* The attack assumes no privileged access and is executed only through queries, making it more realistic than prior works.
* The introduction of bridging steps and progressive shortening demonstrates careful and creative thinking in constructing the attack.
* The paper evaluates MINJA on multiple agents and datasets, showing consistent effectiveness and generalization.
* The paper is well-written and easy to follow, even for readers not deeply familiar with LLM agent systems.

Weakenss:
* A key assumption in the paper is that the LLM agent uses a shared memory bank across users. However, in many real-world applications (such as ChatGPT) each user typically has a private memory that is not accessible or shared without explicit permission. This limits the direct applicability of the proposed attack. The authors acknowledge this assumption and justify it with examples of shared memory systems, so while it is a valid limitation, it may be more relevant in future or specialized deployments.
* The attacker is assumed to be able to inject a relatively large number of interaction records into the system, which may not always be feasible, especially in high-security or rate-limited environments. This constraint is worth discussing more in terms of attack scalability.
* The paper briefly discusses possible defenses (like prompt-level detection), but more detailed analysis of practical defense mechanisms, such as memory isolation, permission control, or system-level safeguards, is lacking. This would help better assess the practical risk and how MINJA can be mitigated in real-world systems.
* The success of the attack depends on how the agent retrieves memory records (e.g., based on query similarity). While the paper explores several embedding models, if real-world systems use stricter or more diverse filtering methods, the effectiveness of the attack may drop.
* In the method section, both q (italic) and q (regular) are used to denote different queries. Could you consider clarifying or changing the notation? It’s easy to confuse them when reading.

---

> ### Author Rebuttal · Authors · 2025-07-31
>
> Thank you for your thoughtful and generous review!  We deeply appreciate your recognition of the realistic threat model, the careful design of our attack strategy, and the clarity and rigor of our experimental evaluation.
>
> **W1: The attack assumes a shared memory bank across users.**
>
> **R1:** Thank you for this insightful comment!
>
> Our work does consider a setting where the agent uses a shared memory bank, which, as noted in the paper, is “common in existing agent frameworks due to deployment and performance considerations.” (Page 3-4, Lines 142-146) This design supports cross-user knowledge accumulation and is increasingly adopted in real-world systems [1][2][3][4][5].
>
> If the memory is isolated, potential attack strategies remain feasible. For example, adversaries could disguise their identity or hijack user accounts to inject poisoned data into private memory, without requiring designer- or system-level access.
>
> We will add more discussion on guidelines for future and specialized agent deployments. Thanks again for your thoughtful comment!
>
>
>
> **W2: The attack assumes the ability to inject a large number of interaction records, which may not be feasible in high-security or rate-limited environments.**
>
> **R2:** Thank you for this important point!
>
> While our main experiments assume a reasonable volume of injected queries for each attack on a certain victim-target pair, we note that MINJA does not require concentrated injection from a single entity. Instead, multiple attackers could collaborate or operate independently in parallel—or a single attacker could use multiple accounts—to collectively contribute malicious records over time.
> This distributed injection strategy mitigates the effect of rate limiting and aligns with realistic adversarial behaviors, such as the use of botnets or compromised user accounts. This strategy helps bypass rate-limited environments, making the attack more feasible and covert in practical settings.
>
> We will add the discussion in the revision.
>
>
>
> **W3: The paper lacks detailed analysis of practical defense mechanisms (e.g., memory isolation, permission control, system-level safeguards), which would help assess real-world risk and mitigation of MINJA.**
>
> **R3:** Thank you for the thoughtful suggestion!
>
> Regarding memory isolation, while it can reduce cross-user poisoning, we note that identity disguise strategies, such as account hijacking or impersonation, remain feasible and have been observed in real-world attacks. These strategies allow adversaries to inject malicious records even under isolated settings.
>
> For permission control, our threat model assumes that attackers have the same interface and capabilities as benign users without any privileged access (Page 3, Lines 138-141). Since attackers are not explicitly distinguishable from regular users, applying strict permission constraints may unintentionally impact benign functionality.
>
> As for system-level safeguards (e.g., GuardAgent [6] based on safety rules), we’d like to emphasize that our injected reasoning steps are plausible, semantically coherent, and aligned with the task context. Detecting such malicious content likely requires external knowledge bases, structured monitoring, or task-specific constraints. For example, in the case of RAP Agent, determining whether a product is truly out of stock based on the indication prompt may require checking the inventory database, which lies beyond the model’s native capabilities.
>
> We agree that developing such targeted defenses is an important and promising direction for future research. We will include those discussions in *Section 5.4 (Potential Defense)* in our revision, and we hope our work motivates further exploration in this area.
>
> **W4: The attack’s effectiveness may degrade if real-world systems adopt stricter or more diverse memory retrieval filtering beyond embedding-based similarity.**
>
> **R4:** Thank you for the insightful question.
>
> 1) Retrieving memory via query similarity is a scalable and widely adopted strategy across LLM-based agents (e.g., RAP, EHR Agent).
>
> 2) Even if retrieval is not based on embedding similarity—e.g., by directly querying an LLM—our attack remains effective. This is because agents typically reuse prior records as in-context demonstrations. Our PSS strategy ensures that the most informative record (from the previous shortening step) remains highly relevant and thus likely to be selected.
>
> 3) While real-world systems may employ stricter filtering or security checks, we show in Section 5.4 that both embedding-level sanitization and prompt-level defenses are limited: semantic overlap makes it hard to distinguish benign vs. malicious records, and LLM-based filtering either fails to generalize (targeted prompts) or suffers from high false positives (general prompts).
> These findings suggest that MINJA remains effective even under more advanced or stricter memory filtering settings, making it a broadly applicable threat model for memory-based agents.
>
> We will add Point 1. to *Section 2 (Related Work)* to highlight the common use of query similarity-based retrieval, and Point 2. will be included in *Section 5.3 (Ablation Study, Choice of Embedding Model for Memory Retrieval)* to discuss MINJA’s effectiveness under LLM-based retrieval mechanisms in our revision.
>
> **W5: The usage of q (italic) and q (regular) for different queries may confuse.**
>
> **R5:** Thank you so much for your valuable suggestion!
> We will revise the notation for clarity—using ***a*** for attack query in the updated version.
>
> [1] Memory sharing for large language model based agents
>
> [2] Koma: Knowledge-driven multi-agent framework for autonomous driving with large language models
>
> [3] Ehragent: Code empowers large language models for few-shot complex tabular reasoning on electronic health records
>
> [4] A language agent for autonomous driving
>
> [5] Rap: Retrieval-augmented planning with contextual memory for multimodal llm agents
>
> [6] GuardAgent: Safeguard LLM Agents by a Guard Agent via Knowledge-Enabled

---

> > ### Author Response · Authors · 2025-08-08
> > **Grateful for your surpportive comments and Rebuttal follow-up**
> >
> > We sincerely appreciate your supportive and thoughtful comments during the review process! They have been very motivating for our team. We hope our rebuttal has addressed your concerns clearly and thoroughly.
> >
> > If you have a moment, we would be grateful if you could kindly let us know whether there are follow-up questions.
> >
> > Thank you again for your time, constructive feedback, and support for our work!

---

> > > ### Comment · Reviewer_KhxJ · 2025-08-09
> > >
> > > Thanks for the rebuttal. Since most of my concerns have been addressed, I would like to maintain my positive rate.

---

### Official Review · Reviewer_qaRt · 2025-06-28

**Clarity:** 3
**Significance:** 2
**Originality:** 3
**Rating:** 4
**Confidence:** 3

**Summary:**

This paper proposes a novel attack on the memory module of reasoning-enabled LLM agents. The attacker is assumed to lack direct access to the memory bank and can only influence it through interactions with the agent. The objective is to poison the memory such that, when a victim query comes, the agent: (1) interprets the victim’s query (q_v) as a target query (q_t), and (2) generates a response aligned with the attacker’s intent. To achieve this, the authors introduce a “bridging step”—a generic but coherent reasoning step that subtly links the attacker’s query to the target intent. Without direct memory access, the attacker induces the agent to generate these bridging steps autonomously by prepending crafted chain-of-thought reasoning to their queries. To improve stealth and efficiency, the authors propose a Progressive Shortening Strategy, which iteratively reduces the length of the poisoned queries while retaining their effect, ultimately producing prompts that appear benign. Experiments across multiple agents and datasets, along with ablation studies, demonstrate the effectiveness of the attack and provide preliminary evaluations of potential defenses.

**Questions:**

+ How sensitive is the attack to changes in the retriever design? Would switching to a different similarity metric, e.g., switching from cosine similarity to dot product, influence the ASR?

+ What is the minimum retrieval rank at which the poisoned record must appear for the attack to succeed? Does attack success drop off significantly as the poisoned record moves past the top-k?

+ How many interaction rounds does the PSS typically require in practice? Would aggressive rate limiting (e.g., 5 queries/hour) substantially hinder the attack’s feasibility?

+ How effective would a simple defense be that detects and then rejects chain-of-thought or multi-step reasoning from user input? Will the attack still stay effective? Additionally, simple rule-based checks (e.g., enforcing that the primary entity mentioned in the answer matches that in the query) could prevent certain examples, like the medical scenario shown in the paper. How does this defense affect its effectiveness?

**Ethical Concerns:**

["NO or VERY MINOR ethics concerns only"]

**Final Justification:**

The author has added experiments evaluating the robustness of the attack under memory retrieval mechanisms with noise and examined its sensitivity to changes in retriever design. They also clarified key technical details, including the required maximum attack rounds and the potential effectiveness of a CoT-based defense. These additions address my concerns, so I would like to maintain the positive score.

**Limitations:**

Yes.

**Paper Formatting Concerns:**

No.

**Quality:**

3

**Strengths And Weaknesses:**

Strengths:

+ The paper introduces a novel and practical threat model in which the attacker has no access to the memory bank and no direct write access to storage or model weights. This setting closely mirrors realistic deployment constraints.

+ The attack is well-motivated; it leverages agent logging and retrieval mechanisms cleverly, enabling memory poisoning purely through interaction. The bridging sentences are effective in making the injected reasoning logical, increasing the plausibility of the attack.

Weaknesses:

+ The paper would benefit from more extensive evaluation of potential (adaptive) defenses, particularly those that may be deployed in realistic LLM agent systems, see more details in the question section.

+  It remains unclear how robust the attack is when the agent incorporates noise, relevance scoring, or temporal decay in its memory retrieval logic—such factors may affect the long-term impact of poisoning.

---

> ### Author Rebuttal · Authors · 2025-07-31
>
> Thank you for your thoughtful review! We sincerely appreciate your recognition of the novelty of our threat model and the effectiveness of our interaction-based memory poisoning approach.
>
> **W2: Unclear robustness of the attack under memory retrieval mechanisms involving noise, relevance scoring, or temporal decay.**
>
> **R1:** Thank you for your very thoughtful question! We would like to discuss the cases mentioned by the reviewer, respectively.
> 1) *Involving noises.* We conducted an ablation experiment on RAP Agent (with GPT-4o) to evaluate the robustness of our attack under retrieval noise. Specifically, we added Gaussian noise (σ = 0.01) to the embedding vectors during memory retrieval, simulating noisy retrieval. The noise level is comparable to the scale of real embeddings (∼e−2), making it a moderate perturbation. We evaluate on victim-target pairs 1, 4, and 7 and report average results. As shown in the table below, ISR remains 100%, and ASR only slightly drops (97.8 to 95.6), indicating MINJA’s strong robustness to noisy retrieval.
> ||**ISR**|**ASR**|
> |-|-|-|
> |Without noise|100.0|97.8|
> |Gaussian noise (σ = 0.01)|100.0|95.6|
>
> 2) *Relevance scoring.* MINJA is robust to changes in the design of the retrieval mechanism, including both similarity metrics and retriever architectures. As detailed in our paper (Page 8, Lines 325-529), we evaluated six retrievers—DPR, REALM, ANCE, BGE, MiniLM, and ada-002—that differ in embedding models. Across all settings, ISR and ASR remain consistently high, demonstrating the attack’s resilience to different relevance scoring strategies.
>
> 3) *Temporal decay.* Temporal decay typically affects all memory poisoning attacks by diminishing the impact of earlier injected records. But our attack is achieved by the adversarial user's interaction with the agent, which could happen anytime, e.g., just before the victim user's interaction.
>
> We will add these results and the discussion to our revised paper.
>
> **Q1: Unclear sensitivity of the attack to changes in retriever design, such as different similarity metrics (e.g., cosine vs. dot product).**
>
> **R2:** Thank you for raising this point!
>
> To assess the sensitivity of MINJA to retriever design choices, we conducted experiments using six different embedding models for memory retrieval, as described in our paper (Page 8, Lines 325-529): DPR, REALM, ANCE, BGE, MiniLM, and OpenAI's ada-002. These models differ not only in their architecture but also in the similarity metrics typically employed—**some use dot-product (e.g., DPR, REALM), while others use cosine similarity (e.g., MiniLM, BGE).**
>
> Despite these variations, as shown in Figure 3, MINJA consistently achieves strong Injection Success Rate (ISR) and Attack Success Rate (ASR) across all tested models. This indicates that the attack is robust to the choice of similarity metric and retriever architecture, demonstrating its generalizability across a range of retrieval mechanisms.
>
> **Q2: What is the minimum retrieval rank at which the poisoned record must appear for the attack to succeed? Does ASR drop significantly when the poisoned record is retrieved beyond top-k?**
>
> **R3:** Thank you for your insightful question!
>
> We analyzed the retrieval ranks of poisoned records in EHR Agent as an example. In successful attacks on both MIMIC-III and eICU, the poisoned record is **almost always retrieved at rank 1**, whereas failed attacks are typically associated with lower-ranked or unretrieved poisoned records.
>
> As with prior memory poisoning attacks such as AgentPoison, if no poisoned record is retrieved, the attack cannot succeed. Thus, the retrieval rank is indeed important, but this is a shared limitation across memory-based attacks, such as AgentPoison.
> To address this inherent limitation of memory-based attacks, MINJA diversifies the victim queries during poisoning (see Page 5, Lines 184–186), which increases the likelihood that different malicious records are retrieved for different user queries and simultaneously promotes higher retrieval ranks for poisoned records.
>
> **Q3: How many interaction rounds does PSS typically require? Would aggressive rate limiting hinder the attack’s feasibility?**
>
> **R4:** Thank you for your insightful question!
>
> PSS typically requires 4–5 interaction rounds. In our main experiments, the number of PSS iterations varies slightly across settings — 4 for EHRAgent on MIMIC-III, and 5 for EHRAgent on eICU, RAPAgent, and QAAgent. We find that even this moderate number of interactions enables consistently high ISR and ASR.
>
> Regarding rate limiting, we note that attackers operate under the same interface and limitations as any benign user (Page 3, Lines 138-141). If system-wide rate limits are enforced, they affect all users equally, but the attacker can use multiple user accounts to attack.
>
> **W1, Q4: Would defenses, such as rejecting chain-of-thought reasoning or enforcing entity consistency between query and answer, reduce the attack’s effectiveness?**
>
> **R5:** We sincerely appreciate your thoughtful question!
>
> We clarify that CoT (chain-of-thought) reasoning is common in benign user queries. Blocking it may negatively impact normal agent usage. Moreover, our indication prompts are designed to be plausible, making them difficult to distinguish from regular CoT responses to benign inputs.
>
> We also evaluated the rule-based defense, which is conceptually aligned with *enforcing entity consistency*. Both methods aim to detect semantic irregularities introduced by the attacker. As mentioned in Section 5.4, we adopt prompt-level detection with GPT-4o to flag malicious queries. Results show that while the task-specific defense prompt for EHRAgent achieves a high precision (e.g., 131/135 on MIMIC), it fails to generalize to RAP or QA agents. In contrast, a more general prompt improves detection across tasks but leads to substantial false positives (e.g., 34/50 false alarms on EHR-MIMIC). This highlights a core challenge: simple rule-based defenses struggle to balance precision and generality, especially when indication prompts are carefully crafted.

---

> > ### Comment · Reviewer_qaRt · 2025-08-02
> >
> > Thank the author for the rebuttal, and it addresses my concerns.

---

> > > ### Author Response · Authors · 2025-08-03
> > > **Thank you for your feedback and support**
> > >
> > > Thank you for your kind comment! We’re truly grateful that our rebuttal addressed your concerns, and we appreciate the time and thoughtful feedback you provided throughout the review process.

---

### Official Review · Reviewer_Sef1 · 2025-06-29

**Clarity:** 4
**Significance:** 3
**Originality:** 3
**Rating:** 4
**Confidence:** 3

**Summary:**

LLM agents with compromised memory banks can generate harmful outputs when the retrieved past records are malicious. This paper proposes a novel method for injecting such malicious records into the memory bank indirectly, solely through interaction with the agent. Unlike previous works that assume the attacker has direct access to the agent’s memory bank, this approach addresses a more realistic and easily exploitable injection pathway.
The proposed method works by injecting indication prompts that guide the agent to autonomously generate malicious reasoning steps. This process involves scenarios where a victim term in the query is implicitly transitioned into a target term, thereby leading to harmful outcomes.

**Questions:**

- The Utility Drop (UD) appears to be a metric where lower values are better, yet Table 1 labels it as “UD↑”. Could the authors clarify why it is presented this way?

- Is it possible to automate the process of crafting the indication prompt, rather than relying on human design? Additionally, how does the degree of logical implausibility in these bridging steps affect the injection success?

**Ethical Concerns:**

["NO or VERY MINOR ethics concerns only"]

**Final Justification:**

The authors have adequately resolved my concerns.

**Limitations:**

- The requirement for manually crafted bridging steps imposes limitations in terms of attack automation and scalability.

- The assumption that the attack only works when a semantically plausible connection between the victim and target terms exists may not hold in many real-world scenarios, making the method less generalizable in practice.

**Paper Formatting Concerns:**

no formatting concerns

**Quality:**

3

**Strengths And Weaknesses:**

strengths

- In addition to injecting indication prompts that guide the agent to follow malicious reasoning steps, the authors introduce a progressive shortening strategy that gradually removes the prompt. This strategy enhances the likelihood of the victim term being seamlessly replaced by the target term during future queries.

- By eliminating the need for direct access to the memory bank and relying solely on queries to inject malicious records, the proposed method increases the practicality and real-world feasibility of memory poisoning attacks.

- The authors conduct extensive experiments using metrics such as Inject Success Rate (ISR), Attack Success Rate (ASR), and Utility Drop (UD) across various conditions, including different embedding models and prior poisoning scenarios, demonstrating the effectiveness and robustness of the proposed approach.


weaknesses

- While Section 4.2 states that “PSS enables the injection of a greater number of relevant malicious records into the memory bank,” the paper lacks an ablation study directly isolating and quantifying the impact of the progressive shortening strategy on injection performance.

- Although the Inject Success Rate (ISR) is consistently high across experiments, the Attack Success Rate (ASR) shows considerable variance, indicating instability in actual malicious influence over agent outputs.

- While MINJA demonstrates high success rates, its reliance on semantically plausible victim-to-target transitions limits its applicability. The attack assumes that a logically coherent bridging can be constructed between the victim and target terms. This could be an assumption that may not hold in many real-world settings where such transitions are implausible or nonsensical.

- The requirement for human-designed prompt presents a limitation in terms of scalability and automation. This hand-engineered component demands attacker expertise and domain knowledge, making the approach less viable in broader or more complex scenarios.

- The trigger design in MINJA is inherently tied to specific victim terms for which semantically plausible bridging to target terms can be crafted. This dependence limits the flexibility and generalizability of trigger word selection.

---

> ### Author Rebuttal · Authors · 2025-07-31
>
> Thank you for your valuable comments! We sincerely appreciate your recognition of MINJA’s practicality, innovation, and empirical validation.
>
> **W1: Isolating the impact of progressive shortening strategies**
>
> **R1:** Thank you for your insightful comment.
>
> We conducted an ablation study on EHRAgent using e-ICU to assess how reducing the number of shortening steps in the PSS process affects the injection success rate (ISR).
>
> Specifically, we compare our default PSS with fewer shortening steps and no PSS at all on three arbitrarily selected pairs (Pair 1, 4, 7) on EHRAgent for eICU.
>
> | Metric | **Full PSS** | **PSS with fewer steps** | **No PSS** |
> |--------|--------------|---------------------------|------------|
> | ISR    | 93.3         | 80.0 | 75.6       |
> | ASR    | 87.8         | 87.8  | 82.2       |
>
> The results show that reducing the number of shortening steps leads to a decrease in injection success rate (ISR), from 93.3% (full PSS) to 80% (fewer steps), and further down to 75.6% when PSS is removed entirely, which validates that PSS facilitates the injection of more malicious records into the memory bank as we expected.
>
> We will add complete experiments and the results to our revision.
>
>
> **W2: ASR’s high variance indicates instability malicious influence**
>
> **R2:** Thank you for raising this concern.
>
> We would like to clarify that our method mainly focuses on memory injection in realistic settings (see Page 2, Line 73) and, as expected, achieves consistently high ISR across settings. The observed variance in ASR mainly arises across different victim-target pairs, which is often seen in the backdoor literature. For example, [1] (Figure 11 & 12) also reports substantial ASR differences across different physical patterns.
>
> To validate this, we conduct additional evaluations on both RAP and QA agents, using GPT-4 and GPT-4o as core models. We selected three arbitrary victim-target pairs (Pairs 1, 4, and 7), and sampled 18 test queries for each pair (with replacement for QA Agent due to its limited test queries). We repeated this process across three runs (test 1,2,3) and measured the standard deviation of ASR.
>
>
> ***RAP Agent***
>
> |**Victim-target Pair**| **test1**| **test2**| **test3**| **standard deviation**|
> |-|-|-|-|-|
> | Pair 1 (GPT-4)     | 72.2  | 77.8  | 77.8  | 2.64|
> | Pair 4 (GPT-4)     | 50.0  | 55.6  | 55.6  | 2.64|
> | Pair 7 (GPT-4)     | 88.9  | 94.4  | 94.4  | 2.59|
> | Pair 1 (GPT-4o)    | 100.0 | 100.0 | 100.0 | 0.00|
> | Pair 4 (GPT-4o)    | 88.9  | 94.4  | 88.9  | 2.59|
> | Pair 7 (GPT-4o)    | 100.0 | 100.0 | 100.0 | 0.00|
>
> ***QA Agent***
>
> | **Victim-target Pair**| **test1**| **test2**| **test3**| **standard deviation**|
> |-|-|-|-|-|
> | Pair 1 (GPT-4)| 66.7  | 61.1  | 61.1  | 3.21|
> | Pair 4 (GPT-4)| 50.0  | 44.4  | 38.9  | 5.56|
> | Pair 7 (GPT-4)| 55.6  | 50.0  | 44.4  | 5.56|
> | Pair 1 (GPT-4o)| 55.6  | 61.1  | 50.0  | 5.56|
> | Pair 4 (GPT-4o)| 55.6  | 50.0  | 61.1  | 5.56|
> | Pair 7 (GPT-4o)| 72.2  | 61.1  | 66.7  | 5.56|
>
> The results demonstrate that ASR remains stable within each pair (standard deviation lower than 3% in RAP Agent, and lower than 6% in QA Agent).
>
>
> Thanks again for raising this point, and we will add the results to our revision.
>
>
> **W3, L2, Q2:  MINJA’s reliance on semantically plausible victim-target transitions, and how the bridging step affects injection success?**
>
>
> **R3:** Thanks for raising these two concerns! We’ll clarify them separately:
>
> 1) As for MINJA’s reliance on semantically plausible victim-target transitions, in most real-world agent applications, a semantically coherent transition between victim and target is both common and practical. We have demonstrated such applicability for four agent applications: in a healthcare agent (e.g., EHR agent), victim and target terms such as patient IDs (on MIMIC-III) or medication (on e-ICU) names are inherently meaningful and contextually coherent. Similarly, in a shopping agent (e.g., RAP agent), both victim and target entities typically represent product names, which often share strong semantic or categorical relationships. For the QA Agent, we consider terms from specific subjects as the victim.
>
>     In other applications, such as code generation agents (e.g., MetaGPT [2]), financial advisors (e.g., FinGPT [3]), or travel assistants(e.g., TravelPlanner [4]), victim and target terms (e.g., function names or data types, financial terms, destinations) are also semantically meaningful, making coherent bridging both feasible and practical across domains.
>
>     Additionally, MINJA’s reliance on semantically plausible victim-target transitions is not a limitation but a deliberate design choice. In practice, attackers can choose both victim and target terms to ensure logical coherence, making such transitions feasible and effective in real-world scenarios.
>
> 2) To directly show the importance of the bridging step *(indication prompt)* plausibility, we conducted an additional ablation study on RAP Agent using victim-target pairs 1, 4, and 7. In this setting, we replaced the entire indication prompt with the target term itself, thereby removing any semantically plausible bridging content. *Note that an attacker will never do this in practice*. As a result, no bridging steps were generated. This manipulation led to **both ISR and ASR dropping to 0%**, demonstrating that the presence of a logically plausible bridging step is crucial for successful injection.
>
>
> **W4, Q2, L1: The reliance on human-designed prompts limits scalability and requires attacker expertise. (Can the prompt generation be automated?)**
>
> **R4:** Thank you for raising this concern!
>
> Our prompt generation does not demand attacker expertise and domain knowledge. In fact, the indication prompts can be generated fully automatically.
> Specifically, we provided the LLM with demonstrations from EHR Agent on MIMIC-III that include bridging steps, and instructed it to generate a logical transition (i.e., an indication prompt) for a fixed victim-target pair in eICU. We then tested the attack using the generated prompt on EHR Agent (eICU) for three pairs (Pair1, 4, 7).
>
> | **Victim-target Pairs**| **ISR**  | **ASR**  |
> |-|-|-|
> | Pair 1 | 80%   | 50%   |
> | Pair 4| 60%   | 60%   |
> | Pair 7| 53.3% | 50%   |
>
> Despite being generated by the model without any manual engineering, the prompts still retained some effectiveness, suggesting that our method supports potential automation and scalability. Note that this is merely a ***naive*** prompting strategy, and a more sophisticated automatic pipeline could likely yield even better results.
>
> **W5: The attack relies on victim terms that allow plausible bridging, limiting trigger selection flexibility.**
>
> **R5:** Thank you for raising this concern!
>
> 1) We would like to clarify that, in our setting, the attacker’s goal is to **influence model behavior toward a specific victim term**. Thus, the victim itself can be regarded as an implicit *trigger* instead of any *explicit triggers* for traditional backdoor attacks.
>
>     Moreover, our victim selection represents a practical improvement over prior methods like BadChain, which rely on injecting explicit triggers into user prompts—an unrealistic assumption. In contrast, MINJA operates without access to user inputs and instead performs memory injection, better aligning with real-world threat scenarios.
> 2) Additionally, as discussed in (W3, L2, Q2) point 1., our framework is designed to be flexible, allowing the attacker to arbitrarily choose the victim term depending on their intent. For instance, in RAP Agent, we target product names, while in EHR Agent, we use entities such as patient IDs or medications. These choices demonstrate that our method can adapt to a wide range of victim-target pair selections across domains.
>
> **Q1: Table 1 presents Utility Drop (UD) with an upward arrow (“UD↑”),**
>
> **R6:** We thank the reviewer for pointing this out! UD should have been labeled as **“UD↓ ”**, indicating that smaller utility drop values are preferred. We will fix it in our revision.
>
> [1] Targeted Backdoor Attacks on Deep Learning Systems Using Data Poisoning
>
> [2] MetaGPT: Meta Programming for A Multi-Agent Collaborative Framework
>
> [3] FinGPT: Open-Source Financial Large Language Models
>
> [4] TravelPlanner: A Benchmark for Real-World Planning with Language Agents

---

> > ### Comment · Reviewer_Sef1 · 2025-08-04
> >
> > The believe the authors have resolved my concerns, and I will reflect this on the review.

---

> > > ### Author Response · Authors · 2025-08-04
> > > **Thank you for your supportive feedback**
> > >
> > > Thank you very much for your kind follow-up and for taking the time to review our rebuttal! We truly appreciate your thoughtful comments and are glad to hear that your concerns have been resolved.

---

### Decision · Program_Chairs · 2025-09-17

**Decision:**

Accept (poster)

**Comment:**

This paper presents a novel memory injection attack against LLM agents that operates solely through query-based interactions rather than requiring direct memory access. The work addresses a gap in existing research by operating through a more realistic threat model.

**Strengths:** Unlike prior work, the attack requires no privileged access to memory banks, making it more realistic for real-world deployment. The experimental validation across multiple agents, datasets, and metrics (ISR, ASR, UD) shows consistently high injection success rates.

**Weaknesses:** All reviewers noted inadequate exploration of realistic defense mechanisms and adaptive countermeasures that would likely exist in production systems. The attack also requires substantial interaction volume and human-crafted prompts, limiting automation and broad deployment.

Overall, the paper makes a valuable contribution by addressing a more realistic attack scenario and demonstrating technical feasibility. During the rebuttal, all reviewers indicated that most of their concerns were successfully addressed.